# Application of Original Therapy for Stimulation of Oral Areas Innervated by the Trigeminal Nerve in a Child with Beckwith–Wiedemann Syndrome

**DOI:** 10.3390/brainsci13050829

**Published:** 2023-05-21

**Authors:** Wioletta Pawlukowska, Michał Patalan, Ewelina Bagińska, Maria Giżewska, Marta Masztalewicz

**Affiliations:** 1Department of Neurology, Pomeranian Medical University, 71-252 Szczecin, Poland; baginska_e@o2.pl (E.B.); marta.masztalewicz@pum.edu.pl (M.M.); 2Department of Pediatrics, Endocrinology, Diabetology, Metabolic Diseases and Cardiology, Pomeranian Medical University, 70-204 Szczecin, Poland; michal.patalan@pum.edu.pl (M.P.); maria.gizewska@gmail.com (M.G.)

**Keywords:** Beckwith–Wiedemann syndrome, speech therapy, oral cavity, stimulation

## Abstract

About 90% of children diagnosed with classic BWS have macroglossia, and 40% of them are submitted to surgical tongue reduction. The purpose of our article is to present a case study of a 5-month-old child with BWS who was treated with an original therapy for stimulation of oral areas innervated by the trigeminal nerve. The therapy included stimulation of the upper and lower lip and muscles of the floor of the mouth. The treatment was provided by a therapist once a week. In addition, the child was stimulated every day at home by his mother. After 3 months, a significant improvement in oral alignment and function was achieved. Preliminary observations of therapy application for stimulation regions innervated by the trigeminal nerve in children with Beckwith–Wiedemann syndrome seem promising. The original therapy for stimulation of oral areas innervated by the trigeminal nerve is a good alternative to existing methods of surgical tongue reduction in children with BWS and macroglossia.

## 1. Introduction

Beckwith–Wiedemann syndrome (BWS, MIM #130650) is a rare (1:10,000–1:15,000 live births) genetically determined congenital growth disorder. It is usually caused by genomic imprinting disturbances (mainly DNA methylation abnormalities), which affect the expression of a few important growth-regulating genes localized on chromosome 11p15, such as *CDKN1C* and *IGF2* [1,2]. The majority of cases occur sporadically (85%) [3].

Typical clinical signs of BWS include macroglossia, macrosomia, lateralized overgrowth, visceromegaly, abdominal wall defects, naevus simplex, and characteristic dysmorphic features such as anterior linear ear lobe creases and/or posterior helical ear pits. Almost half of all newborns with BWS develop hyperinsulinemic hypoglycemia, which can lead to developmental disorders [4]. Children with BWS have an increased risk for embryonal tumors, such as Wilms tumor or hepatoblastoma. The majority of phenotypic features are most pronounced in the first years of life, and the diagnosis of the syndrome is usually established during this period [4,5,6,7,8,9].

The most common symptom of BWS is macroglossia, which is defined as an abnormally large tongue that protrudes beyond the line of teeth and gums in the resting position [4,5]. BWS is believed to be the most common cause of tongue overgrowth. Macroglossia occurs in about 90% of patients with BWS, affects all three dimensions of the tongue, and is usually visible immediately after birth. In some cases, only hemihyperplasia of the tongue is observed [10]. Macroglossia in some patients might result in food swallowing disorders, drooling, abnormal speech development, breathing difficulties (especially when lying down or eating), and even sleep apnea conducted by obstruction of the upper respiratory tract. The psychological aspect should also be taken into consideration, as severe macroglossia, with speech disorders and drooling, can create a false image of a person with an intellectual disability, leading to a lack of self-acceptance and to difficulties in establishing relationships with others. Patients are often referred for intensive speech therapy. The tongue, stacked on teeth, frequently leads to the formation of malocclusion, requiring orthodontic interventions [11,12,13]. As the child grows, tongue growth decreases, the mandible enlarges, and oral cavity volume can offset the dysmorphia [12,13,14]. In cases of significantly severe macroglossia, surgical tongue reduction should be considered. Previous studies indicate that about 40% of patients with BWS require surgery. It is usually performed at 2–3 years of life [14]. Some surgeons recommend performing the procedure not earlier than 6 months old, taking into consideration the increased risk of macroglossia recurrence in very young patients submitted to surgery [15]. In rare cases, recurrence of macroglossia after surgery is observed even in older children. Although long-term complications occur rarely, patients have been described as struggling with post-operative complications of tongue mobility, abnormalities of its shape, and taste disorders [16]. Particularly for the group of the youngest children, it may be beneficial to look for alternative methods of treatment, which have not been described so far.

In this article, we present the original therapy for stimulation of oral areas innervated by the trigeminal nerve, improving the alignment of the articulatory organs of the tongue, lips, and mandible and reducing the risk of macroglossia complications in children with BWS.

## 2. Case Presentation

This case is of a child from a second pregnancy (in vitro), second birth terminated by cesarean section due to placenta previa and fetal macrosomia, born at term (38 weeks and 6 days) in good condition (Apgar score of 10), with a birthweight of 3990 g and body length of 56 cm (90–97 percentile). The pregnancy was complicated by the mother’s hypothyroidism, treated with L-thyroxine 75 µg, and at 30 weeks by gestational bleeding (hospitalized in the Department of Pregnancy Pathology). In the first days of life, the baby had temporary difficulties with sucking resulting from abnormal suction of the nipple. The child was primarily breastfed on request, and periodically bottle-fed.

After delivery, the child was diagnosed with macrosomia and dysmorphia (smooth philtrum, wide nasal bridge, and a large, asymmetrical tongue bigger on the left).

At 3 months of age, the final diagnosis of Beckwith–Wiedemann syndrome was established (hypomethylation in KCNQ1OT:TSS:DMR (IC2) locus). In addition, a heart defect in the form of PFO/ASD II was found.

### Speech Pathology Assessment and Intervention

At 5 weeks of life, the child received speech therapy. The conducted assessment revealed a weakness of the orbiticularis oris muscle (Latin: musculi orbiticularis oris pars labialis), smoothing of the nasolabial fold, bilateral drooping of the corners of the mouth, weakening of the rooting reflex, and hypertension in the muscles of the temporomandibular joint. The child’s large tongue was asymmetrical in the alignment in front of the oral cavity during abduction and adduction of the temporomandibular joint. The baby had hypertension in the muscles of the floor of the mouth, excessive drooling, exaggerated licking and palatal and pharyngeal reflexes, and difficulty in transferring a food bolus from the front of the mouth toward the back. During breastfeeding, a lack of proper sealing of the oral cavity generated correct sucking pressure. The child mainly swallowed food during the first phase of feeding and rarely progressed to the active sucking phase, frequently choked, and had large breath fatigue that was predominant while eating. He was fed on demand every 1.5 h.

The child was treated with the original speech therapy based on stimulation of areas innervated by the trigeminal nerve in the oral cavity (Figure 1). Due to exaggerated pharyngeal and gag reflexes, therapy was divided into two phases. Stimulation of the upper lip and mandible was started, and after the vomiting reflex was reduced, the muscles of the floor of the mouth and the temporomandibular joint were stimulated.

The first phase involved stimulation of the upper lip and the chin reflex in the inner and in the outer part of the oral cavity. Therapy began with stimulation of the inner part of the upper lip by massaging in a circular motion towards the nose in the area of the upper lip frenulum. This stimulation allowed the mouth to open and then the tongue to retract into the mouth. Subsequently, the same stimulation was performed on the outer part of the upper lip. Pulling it downward, pressing and holding the upper lip against the lip redness resulted in the closure of the oral cavity. The mandibular reflex area consisted of pressing the chin inward while keeping a slightly open mouth. This movement allowed better joint abduction. The next step was stimulation of the floor of the mouth, starting with a genioglossus muscle (Latin: musculus genioglossus) and going toward the base of the tongue. After therapy of this area, a reduction in the weight of the tongue and a retreat of the tongue into the oral cavity was observed. This stimulation was performed for 1 h once a week. Stimulation of the outer part of the oral cavity was continued at home by the mother for 10 min every day.

After 3 months of therapy, significant improvement in function and alignment of articulatory organs was obtained (Figure 2). Drooling resigned. The child had his mouth closed most of the time, with his tongue retracted. There was a reduction in respiratory effort during eating.

## 3. Discussion

A total of 90% of children diagnosed with classic BWS have macroglossia, and BWS is the most common cause of macroglossia in childhood.

In some cases, macroglossia regresses spontaneously (due to a combination of slowed tongue growth and accelerated mandibular growth), but about 40% of patients are submitted to surgical tongue reduction. The tongue is typically enlarged in all three dimensions, and the goal of the surgery is to improve its function by reducing its volume while maintaining its proper shape. Many surgical techniques are used, but anterior wedge resection is among them the most common. Surgical complications, although rare, may include post-surgical tongue swelling and wound spread [11,13,14].

The child in our care was treated with the original therapy for oral cavity stimulation. The stimulated areas were innervated by the trigeminal nerve. The trigeminal nerve is the largest of all cranial nerves in the face and is a sensorimotor nerve that provides sensory innervation to the face in terms of three main segments (ocular (V1), maxillary (V2), and mandibular (V3)). The motor part innervates the muscles of mastication through the mandibular segment. Branch V1 of a nerve provides sensory innervation to the eye, nose, scalp, and forehead. Branch V2 of a nerve is purely sensory and provides innervation to the maxilla, palate, upper lip, and cheek. Branch V3 of the nerve is the largest sensorimotor branch, which provides sensory information from the lower third of the face, tongue, floor of the mouth, and mandibular, while the motor part innervates the muscles of mastication (temporalis, masseters, and pterygoid), the mylohyoid muscle, and the anterior belly of digastric muscle [17].

Our therapy was based on stimulation of the main reflex sites of the upper and lower lip and the muscles of the floor of the mouth. All of these muscles are innervated by the trigeminal nerve. Stimulation began with the upper lip. This allowed for better control of the tongue as well as the mandible and lip.

Previous studies have shown that the lip shows great variation in excitation, and so stimulation of the corner of the mouth on the same side, chin, and cheek, produces no responses, while stimulation of the edges of the lip vermilion on the same side, produces significant perioral reflex. The results suggest that the upper and lower divisions of the orbicularis oris share a common synaptic drive, at least from the input generated by the reflex pathways, and that the upper lip may be more innervated by mechanoreceptors than the lower lip [18].

This mechanism was used during the stimulation of the upper lip. Our observations show that stimulation (pressure) of the outer part of the upper lip against the lip vermilion causes the corners of the mouth to pull up and close the oral cavity. Stimulation of the upper lip frenulum toward the nose (circular movements) causes the mouth to open and close with retraction of the tongue. This may be due to the fact that the oro-facial sensory part includes significant populations of slow and fast-adapting mechanoreceptors [19,20,21]. These rapidly conducting primary afferent fibers (Aβ) provide the central nervous system with information regarding the sensation of movement and proprioception in the perioral tissues, which is very important, given the apparent lack of supplying muscle spindles. Trigeminal-facial reflexes are thought to fulfill an important role in oral-motor coordination and provide the nervous system with a set of basic coordination patterns that can be activated by sensory stimuli and/or descending signals from the cortex. Most brainstem reflexes, including the perioral reflex, have relatively complex peripheries (compared to the stretch reflex) with the involvement of one or more interneurons and tend to act on orofacial muscle groups [22].

It has also been observed in studies that stimulation of the upper lip mucosa causes contraction and closure of the oral cavity (reflex responses of human perioral muscles to mechanical stimulation of the vestibular oris mucosa). Furthermore, it has been shown that these sensorimotor features encoded by afferent trigeminal nerves are important for motor learning and maintenance of oromotor control during speech, sucking, chewing, swallowing, and gestures [23]. Studies have shown that the masseter, tongue, and lips are functionally interconnected (reflex responses in human jaw, lip, and tongue muscles elicited by mechanical stimulation). Morphological studies have shown interconnections between the red nucleus and the following areas: contralateral, medullary nucleus of the oral trigeminal nerve (Vo); bilateral, medullary trigeminal interporalis nucleus (Vi); and trigeminal principal sensory nucleus [24]. These areas are relay stations that transmit somatosensory information from the oro-facial areas, including nociceptive signals. Responses of somatosensory neurons in the Vo have been found to be modulated by electrical stimulation of the RN [25]. These observations suggest that the RN is involved in the modulation of somatosensory transmission, including nociceptive information, from oral-facial areas [26].

Neuromodulation is a natural physiological process involving changes in neuronal and synaptic properties by neurons or substances released by neurons [27]. Non-invasive stimulation of the trigeminal nerve, as well as non-invasive stimulation of the vagus nerve, should also be considered scientifically validated non-invasive brain modulation techniques [28]. Neuromodulation can modify the intrinsic properties of the neuronal membrane by altering the cellular and/or synaptic properties of neurons. Neuromodulatory stimuli can cause the development of new circuits or sometimes effectively transform old circuits. External modifications can be used not only to regulate one area, but also to organize a system of CNS-regulated circuits [29,30]. Studies have shown that combining transcutaneous ear VNS (taVNS) with bottle-feeding rehabilitation improves oromotor coordination and leads to improved food intake, even in premature infants. EMG electrodes placed on target oro-facial muscles can effectively trigger taVNS stimuli in infants in a closed manner. The vagus nerve can be stimulated non-invasively via the auricular branch. Transcutaneous vagus nerve stimulation appears to be a promising therapeutic option for treating chronic fatigue syndrome. Vagus nerve stimulation (VNS) has been found to promote motor activity by increasing neuroplasticity and facilitating cortical reorganization and neurogenesis [31,32]. Therefore, it seems reasonable to expand the knowledge of cranial nerve stimulation in Beckwith–Wiedemann syndrome, including stimulation of the trigeminal nerve as well as in the future of other nerves, including the vagus nerve, which can improve not only articulatory performance but also hyperinsulinemic hypoglycemia, breathing difficulties, and psychological issues in later life [33,34].

Considering how the trigeminal nerve influences oral control, it seems reasonable to stimulate the areas of the oral cavity supplied by this nerve when the articulatory organs of the tongue, lips, and mandible are misaligned. The original therapy of stimulation of oral areas supplied by the trigeminal nerve is a promising alternative to surgical treatment in tongue hypertrophy in children with BWS.

## 4. Conclusions

Preliminary observations of the application of therapy for stimulation of areas innervated by the trigeminal nerve in children with Beckwith–Wiedemann syndrome seem promising. The original therapy for stimulation of oral regions innervated by the trigeminal nerve is a good alternative to previously used methods of reduction of the tongue in children with BWS and macroglossia.

## Figures and Tables

**Figure 1 brainsci-13-00829-f001:**
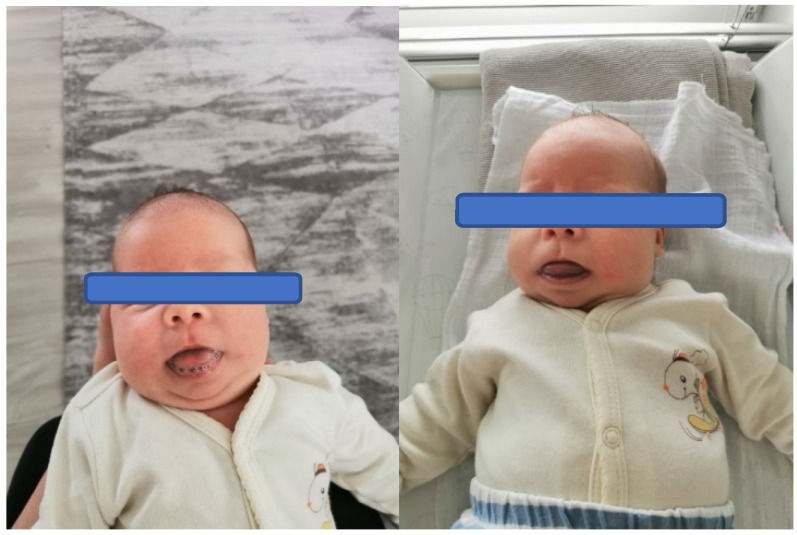
Condition of articulatory organs in a child with Beckwith–Wiedemann syndrome at 6 weeks of age before application of original therapy for stimulation of areas innervated by the trigeminal nerve.

**Figure 2 brainsci-13-00829-f002:**
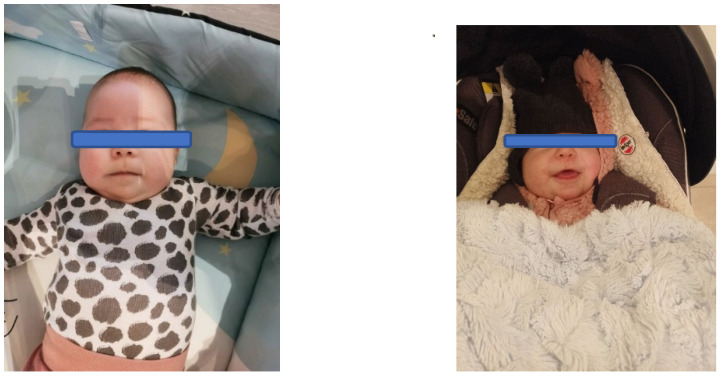
Condition of articulatory organs in a child with Beckwith–Wiedemann syndrome after 3 months of original therapy for stimulation of areas innervated by the trigeminal nerve.

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
