# Peer review of "Application of Original Therapy for Stimulation of Oral Areas Innervated by the Trigeminal Nerve in a Child with Beckwith–Wiedemann Syndrome"

_brainsci, 2023, doi:10.3390/brainsci13050829_

Round 1

Reviewer 1 Report

Very interesting paper, the limit is that just one case is described. My suggestion, is, if possible, to describe a case series of patients treated in the same way.

Author Response

                                                                                        Szczecin, 17nd May 2023

Journal of Brain Sciences

Editors

Manuscript number: brainsci-2397641

Application of the original therapy for stimulation of oral areas innervated by the trigeminal nerve in a child with Beckwith-Wiedemann syndrome by Wioletta Pawlukowska, Michał Patalan, Ewelina Bagińska, Maria Giżewska, Marta Masztalewicz

We were pleased to read the constructive comments of the Reviewers and their suggestion that the manuscript could be considered for publication in the Journal, with some major revision. We reworked and corrected our paper according to the Reviewers’ requests, performing changes in the main manuscript. In the revised manuscript, all changes are indicated using the editing tools. In response to Reviewers, all the changes are indicated in red. We trust that the revised version of our manuscript is clearer and strengthened scientifically. We thank the Reviewers for their comments and careful evaluation of our paper. We are happy to address all the Reviewers’ comments point by point below.

We hope it will now meet with your approval for publication in the Journal of Brain Sciences. Thank you for your time and I am looking forward to hearing from you.

Sincerely yours,

Wioletta Pawlukowska Ph D

Department of Neurology,

Pomeranian Medical University in Szczecin

Unii Lubelskiej 1
71-252 Szczecin, Poland

phone: +48 914800914

fax: +48 914800918

-------------------------------------------Reviewer #1’s Comments -------------------------------------

Dear Reviewer,

Thank you for your comments and kind opinion concerning our manuscript entitled “Application of the original therapy for stimulation of oral areas innervated by the trigeminal nerve in a child with Beckwith-Wiedemann syndrome.

Points of criticism:

My suggestion, is, if possible, to describe a case series of patients treated in the same way.”

(The response)

We realize that the case we have described is only an introduction to further research on the application of the trigeminal nerve stimulation method in children with Beckwith-Wiedemann syndrome, and we hope to present many more such cases in the future.

Reviewer 2 Report

This case reports underlines that original therapy for oral cavity stimulation is easy to implement and can be very helpful to manage macroglossia in Beckwith-Widermann syndrom, at least before 6 months of age (no surgery of the tongue is advised during this period).

Nevertheless, in the discussion of this manuscript, non-invasive trigeminal stimulation as well as noninvasive vagus nerve should also be considered as scientifically validated non-invasive bottom-up brain modulation techniques, easily implemented from the outer ear. (Mercante et al, 2018. Auricular Neuromodulation: The Emerging Concept beyond the Stimulation of Vagus and Trigeminal Nerves. Medicines (Basel). 2018 Jan 21;5(1):10. doi: 10.3390/medicines5010010).

For instance, pairing transcutaneous auricular VNS (taVNS) with bottle-feeding rehabilitation was shown to improve oromotor coordination and lead to improved oral intake volumes, even in premature infants (Badran et al, 2020. Transcutaneous Auricular Vagus Nerve Stimulation-Paired Rehabilitation for Oromotor Feeding Problems in Newborns: An Open-Label Pilot Study. Front Hum Neurosci. 2020 Mar 18;14:77. doi: 10.3389/fnhum.2020.00077.)

Interestingly, EMG electrodes placed on target orofacial muscles can effectively trigger taVNS stimuli in infants in a closed loop fashion. The active lead was placed over the buccinator muscle. The frontal reference lead produced the highest sensitivity.  

(Cook et al, 2020. Design and validation of a closed-loop, motor-activated auricular vagus nerve stimulation (MAAVNS) system for neurorehabilitation. Brain Stimul. 2020 May-Jun;13(3):800-803. doi: 10.1016/j.brs.2020.02.028.)

As a consequence, pairing taVNS with oral cavity stimulation does not bother the oral cavity stimulation since the device is placed on the ear.

These non-invasive techniques can be done at home easily by the mother (Badran et al, 2022. A pilot randomized controlled trial of supervised, at-home, self-administered transcutaneous auricular vagus nerve stimulation (taVNS) to manage long COVID symptoms. Bioelectron Med. 2022 Aug 25;8(1):13. doi: 10.1186/s42234-022-00094-yp.

At last, taVNS could also help other Beckwith-Wiedemann symptoms (hyperinsulinic hypoglycemia, breathing difficulties and psychological issues later in life (for a review, Rangon and Niezgoda, 2022. Understanding the Pivotal Role of the Vagus Nerve in Health from Pandemics. Bioengineering (Basel). 2022 Jul 29;9(8):352. doi: 10.3390/bioengineering9080352.

Author Response

                                                                                   Szczecin, 17nd May 2023

Journal of Brain Sciences

Editors

Manuscript number: brainsci-2397641

Application of the original therapy for stimulation of oral areas innervated by the trigeminal nerve in a child with Beckwith-Wiedemann syndrome by Wioletta Pawlukowska, Michał Patalan, Ewelina Bagińska, Maria Giżewska, Marta Masztalewicz

We were pleased to read the constructive comments of the Reviewers and their suggestion that the manuscript could be considered for publication in the Journal, with some major revision. We reworked and corrected our paper according to the Reviewers’ requests, performing changes in the main manuscript. In the revised manuscript, all changes are indicated using the editing tools. In response to Reviewers, all the changes are indicated in red. We trust that the revised version of our manuscript is clearer and strengthened scientifically. We thank the Reviewers for their comments and careful evaluation of our paper. We are happy to address all the Reviewers’ comments point by point below.

We hope it will now meet with your approval for publication in the Journal of Brain Sciences. Thank you for your time and I am looking forward to hearing from you.

Sincerely yours,

Wioletta Pawlukowska Ph D

Department of Neurology,

Pomeranian Medical University in Szczecin

Unii Lubelskiej 1
71-252 Szczecin, Poland

phone: +48 914800914

fax: +48 914800918

-------------------------------------------Reviewer #2’s Comments -------------------------------------

Dear Reviewer,

Thank you for your comments and kind opinion concerning our manuscript entitled “Application of the original therapy for stimulation of oral areas innervated by the trigeminal nerve in a child with Beckwith-Wiedemann syndrome”. We have studied the comments carefully and have made corrections, which we hope, will meet with your approval.

Points of criticism:

Nevertheless, in the discussion of this manuscript, non-invasive trigeminal stimulation as well as noninvasive vagus nerve should also be considered as scientifically validated non-invasive bottom-up brain modulation techniques, easily implemented from the outer ear. (Mercante et al, 2018. Auricular Neuromodulation: The Emerging Concept beyond the Stimulation of Vagus and Trigeminal Nerves. Medicines (Basel). 2018 Jan 21;5(1):10. doi: 10.3390/medicines5010010).

(The response)

We have completed the missing citation

Non-invasive stimulation of the trigeminal nerve, as well as non-invasive stimulation of the vagus nerve, should also be considered scientifically validated non-invasive brain modulation techniques [29].

Points of criticism:

For instance, pairing transcutaneous auricular VNS (taVNS) with bottle-feeding rehabilitation was shown to improve oromotor coordination and lead to improved oral intake volumes, even in premature infants (Badran et al, 2020. Transcutaneous Auricular Vagus Nerve Stimulation-Paired Rehabilitation for Oromotor Feeding Problems in Newborns: An Open-Label Pilot Study. Front Hum Neurosci. 2020 Mar 18;14:77. doi: 10.3389/fnhum.2020.00077.)

(The response)

We have completed the missing citation

Points of criticism:

Interestingly, EMG electrodes placed on target orofacial muscles can effectively trigger taVNS stimuli in infants in a closed loop fashion. The active lead was placed over the buccinator muscle. The frontal reference lead produced the highest sensitivity.  

(Cook et al, 2020. Design and validation of a closed-loop, motor-activated auricular vagus nerve stimulation (MAAVNS) system for neurorehabilitation. Brain Stimul. 2020 May-Jun;13(3):800-803. doi: 10.1016/j.brs.2020.02.028.)

(The response)

We have completed the missing citation

Points of criticism:

As a consequence, pairing taVNS with oral cavity stimulation does not bother the oral cavity stimulation since the device is placed on the ear.These non-invasive techniques can be done at home easily by the mother (Badran et al, 2022. A pilot randomized controlled trial of supervised, at-home, self-administered transcutaneous auricular vagus nerve stimulation (taVNS) to manage long COVID symptoms.Bioelectron Med. 2022 Aug 25;8(1):13. doi: 10.1186/s42234-022-00094-yp.

(The response)

We have completed the missing citation

Studies have shown that combining transcutaneous ear VNS (taVNS) with bottle-feeding rehabilitation improves oromotor coordination and leads to improved food intake, even in premature infants. EMG electrodes placed on target oro-facial muscles can effectively trigger taVNS stimuli in infants in a closed manner. The vagus nerve can be stimulated non-invasively via the auricular branch. Transcutaneous vagus nerve stimulation appears to be a promising therapeutic option for treating chronic fatigue syndrome. Vagus nerve stimulation (VNS) has been found to promote motor activity by increasing neuroplasticity and facilitating cortical reorganization and neurogenesis [32-33].

Points of criticism:

At last, taVNS could also help other Beckwith-Wiedemann symptoms (hyperinsulinic hypoglycemia, breathing difficulties and psychological issues later in life (for a review, Rangon and Niezgoda, 2022. Understanding the Pivotal Role of the Vagus Nerve in Health from Pandemics. Bioengineering (Basel). 2022 Jul 29;9(8):352. doi: 10.3390/bioengineering9080352.

(The response)

We have completed the missing citation

Therefore, it seems reasonable to expand the knowledge of cranial nerve stimulation in Beckwith-Wiedemann syndrome, including stimulation of the trigeminal nerve as well as in the future of other nerves including the vagus nerve, which can improve not only articulatory performance, but also hyperinsulinemic hypoglycemia, breathing difficulties and psychological issues in later life [34-35]
